# Simulation Study of the Effect of Antimicrobial Peptide Associations on the Mechanism of Action with Bacterial and Eukaryotic Membranes

**DOI:** 10.3390/membranes12090891

**Published:** 2022-09-16

**Authors:** Matko Maleš, Larisa Zoranić

**Affiliations:** 1Faculty of Maritime Studies, University of Split, 21000 Split, Croatia; 2Department of Physics, Faculty of Science, University of Split, 21000 Split, Croatia

**Keywords:** α-helical peptides, antimicrobial peptides, mode of action, molecular dynamic simulations, membrane-peptide interaction, aggregation

## Abstract

Antimicrobial peptides (AMPs) can be directed to specific membranes based on differences in lipid composition. In this study, we performed atomistic and coarse-grained simulations of different numbers of the designed AMP adepantin-1 with a eukaryotic membrane, cytoplasmic Gram-positive and Gram-negative membranes, and an outer Gram-negative membrane. At the core of adepantin-1’s behavior is its amphipathic α-helical structure, which was implemented in its design. The amphipathic structure promotes rapid self-association of peptide in water or upon binding to bacterial membranes. Aggregates initially make contact with the membrane via positively charged residues, but with insertion, the hydrophobic residues are exposed to the membrane’s hydrophobic core. This adaptation alters the aggregate’s stability, causing the peptides to diffuse in the polar region of the membrane, mostly remaining as a single peptide or pairing up to form an antiparallel dimer. Thus, the aggregate’s proposed role is to aid in positioning the peptide into a favorable conformation for insertion. Simulations revealed the molecular basics of adepantin-1 binding to various membranes, and highlighted peptide aggregation as an important factor. These findings contribute to the development of novel anti-infective agents to combat the rapidly growing problem of bacterial resistance to antibiotics.

## 1. Introduction

Biological membranes maintain integrity and support cellular processes [1]. Their main functions include separating cell or internal organelles and other compartments from their surroundings, controlling chemical transport across the membrane, establishing and maintaining transmembrane solute gradients, containing a variety of receptors, cell-cell recognition and adhesion, intercellular and intracellular communication, and energy transduction [2]. Antimicrobial peptides (AMPs), which are activated as part of the cell’s innate defense system against invading pathogens, also have a membrane-related mechanism of action [3]. 

AMPs are found in almost all organisms, and act primarily on evolutionary conserved bacterial membranes [4], making them less likely to induce bacterial resistance [5]. They are used for these reasons in the development of new anti-infective agents, which may be solutions to the rapid emergence of drug-resistant bacteria [6,7]. 

Nowadays it is recognized that the role of AMPs has broad functionality [8,9,10]. Bactericidal activities primarily involve membrane destabilization, the formation of pores or other types of lesions [11], but also include intracellular targets, resulting in the inhibition of nucleic acid or protein synthesis or processes such as cell division and the release of cell wall lytic enzymes [12,13]. Furthermore, AMPs have a broad activity spectrum that includes Gram-negative and Gram-positive bacteria, as well as multidrug-resistant strains [10,14], but they also show efficacy against antimicrobial infections caused by biofilms [15]. They have recently been studied as anticancer [16], antiviral and antifungal agents [7].

AMPs exhibit high variability in sequence and structure but usually share amphipathicity and positive overall charge, the former of which contributes to interactions with negatively charged bacterial membranes, and the latter in adaptation to a polar and hydrophobic membrane environment. Their properties must be finely balanced to increase antibacterial efficacy while remaining non-toxic to host cells [17,18]. Unraveling these rather complex structure-activity relationships is a challenge. In addition to experimental techniques that measure AMP activity on artificially made membrane systems [19] or directly examine processes in living cells [13], computational methods, mostly molecular dynamics (MD), are used to access detailed peptide-membrane interactions [20,21].

Combining both approaches remains a challenge because the spatial and temporal scales available through standard simulation methods are frequently far from those observed in experiments. Not less important is the availability of simulation models (including force fields) that are particularly relevant to membranes, which are highly heterogeneous systems involving hundreds of different lipids and a wide range of compositions in different cell types (and compartments), as well as proteins, which cover approximately 30% of the membrane area [22]. Nonetheless, the development of membrane models has recently accelerated, progressing from simple single-component systems to multicomponent systems, with more realistic models for a wide range of cell types and organelles [23,24]. Simulations have proven to be extremely useful in determining the functionality of membrane-active antimicrobial peptides, and it has been demonstrated that AMPs have complex and sophisticated mechanisms of action, allowing these peptides to adapt to bacterial counter-reactions to their challenge [4,25]. Several membrane-related mechanisms have been observed, including the carpet mechanism, which causes membrane disruption due to induced surface tensions [26], the barrel-stove model [27], which results in the formation of a hydrophobic pore stabilized by peptides structuring along the pore, and the toroidal pore, which is stabilized by peptides in the pore and/or at the pore rim [28,29]. Other possible AMPs mechanisms, such as depolarization or fusion, electroporation, and phospholipid targeting, are less disruptive [30].

In this article, we investigate interactions of some standard and newly proposed membrane models with the designed antimicrobial peptide adepantin-1 [31,32]. Adepantin-1 (GIGKHVGKALKGLKGLLKGLGES–NH_2_) is a glycine-rich peptide with a length of 23 residues constructed by the sequence-based AMP-Designer algorithm [31]. The AMP-Designer program identifies highly selective antimicrobial peptides by predicting the selectivity index, which is the ratio of toxicity to antibacterial activity. Biological characterization showed that adepantin-1 is highly selective for Gram-negative bacteria, has an exceptionally low hemolytic activity, and is less than 50% homologous to any other natural or synthetic antimicrobial peptide [32].

In order to access the underlying molecular mechanism of the experimentally observed results, as well as how the peptide’s high amphipacity affects its antibacterial activity and toxicity, simulation experiments of one or more adepantin-1 peptides interacting with various types of membrane were performed. The simulation membrane models, simplified representations of real membranes with fewer lipid components and atomistic or coarse-grained descriptions of force fields based on pair-interactions, included Gram-positive cytoplasmic membranes that represent *Staphylococcus aureus* (*S. aureus*), Gram-negative cytoplasmic and outer membranes that represent *Escherichia coli* (*E. coli*), and a zwitterionic membrane model for eukaryotic cells.

## 2. Materials and Methods 

### 2.1. Peptide Design and Biological Activity

Adepantin-1 was created using the AMP-Designer algorithm (http://split.pmfst.hr/split/dserv1/ (accessed on 01 March 2022)) [31]. It is the first of seven designed peptides known as adepantins (Automatically Designed Peptide Antibiotics) reported in [32]. The AMP-Designer algorithm was developed using AMPad database [31] of frog-derived, helical antimicrobial peptides with determined selectivity index SI = HC50/MIC, where HC50 is the peptide concentration required to achieve 50% red blood cell lysis and antimicrobial activity is expressed as the minimal inhibitory concentration (MIC) for bacterial growth. The Designer uses the D-descriptor, which is the cosine of the angle between two sequence moments obtained with different hydrophobicity scales, an amino acid selectivity index, a motif regularity index, and other statistical rules extracted from AMPad database to identify highly selective peptide antibiotics.

Adepantin-1 has seven glycine residues out of a total of twenty-three, the majority of which are in the polar sector. Their function may be dual in that they allow unstructured monomers to pass easily through cell wall components and that they increase the likelihood of transient monomer aggregation and pore formation in the cytoplasmic membrane [33]. Adepantin-1 also contains negatively charged glutamic acid, and it has been suggested that negatively charged Asp/Glu residues, which are present in more than 70% of native amphipathic cationic peptides, influence peptide structuring and dimerization, as well as inhibit hemolysis [34,35].

These favorable properties embedded in the adepantin-1 design result in an excellent antimicrobial activity. Previous testing reported in [31,32], showed adepantin-1 to be highly selective for Gram-negative bacteria, with MIC values 2–4 μM against *E. coli* and 16 μM against *Pseudomonas aeruginosa*, but over 128 μM against *S. aureus*. Adepantin-1 also has a remarkably low hemolytic activity (HC50 > 500 μM). When compared to the other AMPs, adepantin has a very high selectivity index equal to 200.

The measurement of circular dichroism (CD) spectra revealed that adepantin-1 has a random structure in aqueous buffer (calculated helicity is less than 5%) but adapts more helical structuring in the presence of organic solvents such as 50% trifluoroethanol (with 35% of helicity). In contrast, the shape of the spectra, particularly the 208/222 ratio in the presence of anionic large unilamellar vesicles (LUVs) suggested that the peptide may interact with these membranes in an aggregated helical form, which was especially evident with the covalently linked dimers. In the presence of neutral LUVs, both monomeric and dimeric peptides remained largely as random coils, indicating that they did not efficiently insert into this type of membrane [32].

According to membrane permeabilization research, treatment with dimeric adepantin-1 resulted in significantly faster membrane lysis than monomeric adepantins. Therefore, dimerization appears to favor outer membrane passage and cytoplasmic membrane permeabilization, which may be related to their greater potency [32].

### 2.2. Simulation Details

Using Gromacs version 2021.3 (Stockholm, Sweden) [36], all-atom (AA) and coarse-grained (CG) molecular dynamics simulations of antimicrobial peptide adepantin-1 in a water-immersed closed environment comprised of various membranes were carried out.

The AA simulations included the following membrane models: (I) a negatively charged bilayer of 1-palmitoyl-2-oleoyl-sn-glycero-3-phosphoethanolamine (POPE) and 1-palmitoyl-2-oleoyl-sn-glycero-3-phosphatidylglycerol (POPG) lipids in a 3:1 mixing ratio modeling the inner membrane of Gram-negative bacteria [37]; (II) a model of Gram-positive plasma membrane consisting of POPG, lysylphosphatidylglycerol (Lys-PG) and 1,10-palmitoyl-2,20-vacenoyl-cardiolipin (PVCL2) lipids in a 57:38:5 mixing ratio [38]; (III) a zwitterionic 1-palmitoyl-2-oleoyl-sn-glycero-3-phosphatidylcholine (POPC) bilayer representative of a eukaryotic cell, and (IV) a model of the *E. coli* outer membrane [39]. AA simulations included cases with one, two and twelve adepantin-1 peptides. CG simulations were performed with the solvated membrane of Gram-negative bacteria and twelve or twenty-four peptides.

Finally, using mapping CG to AA models, the CG2AA all-atom simulations were performed with either twelve or twenty-four peptides interacting with the Gram-negative inner membrane. The CG to AA procedure, also known as backmapping, was as follows. The final configuration from the CG simulations was used as input in CHARMM-GUI Martini to All-atom Converter [40]. It was verified that CG Martini lipids have corresponding all-atom lipids in the CHARMM36m force field, as well as the same charges, so no additional ions were required. The procedure included the use of the python script backward.py for conversion of peptides [41]. The energy minimization process began with Gromacs 5.0 and continued with the same procedure as for other AA simulations.

A model of outer membrane (OM) of Gram-negative bacteria (Figure 9) was built based on the compositions of *E. coli* bacteria. The inner leaflet contained zwitterionic 1-palmitoyl-2-cis-vaccenic-phosphatidylethanolamine (PVPE) (74%) and anionic 1-palmitoyl-2-cis-vaccenic-phosphatidylglycerol (PVPG) (21%) and PVCL2 (5%) phospholipids. The outer leaflet was composed of anionic (charge −10) lipopolysaccharide (LPS) molecules made of Lipid A (Type1 with 6 acyl chains), core region (K12) bonded to 2 of O-antigens units (smooth) or without them (rough) [39] (see Appendix A). The core region contained neutralizing Ca^2+^ ions, while O-antigens were neutralized with ions from solution [42]. 

Details of the simulations and bilayer compositions are presented in Table 1.

The C-QUARK structure predictor (https://zhanggroup.org/C-QUARK/, accessed on 1 January 2022) was used to obtain models for initial peptide structure [43] predicting the α-helical structure (Appendix A). Models for lipids and lipopolysaccharides (Appendix A) were taken from the CHARMM-GUI database [44]. The CHARMM36m force field [45] and TIP3 water model [46] were used in AA simulations, and the Martini 2.2 force-field for CG models [47]. The initial conformations for AA simulations were prepared using the CHARMM-GUI Membrane Builder [48,49], whereas initial conformations for CG simulations were created using the CHARMM-GUI Martini Bilayer Maker [50,51]. 

Peptide charge was defined as pH 7 considering a charged N-terminal amine but neutral amidated C-terminus. The Martini Bilayer Maker does not provide a variety of peptide terminal possibilities; therefore, we manually amidated the C-terminal by first charging both terminals and then releasing the charge on the C-terminal by swapping the Qa bead for a P5 bead [52]. A water layer of at least 4 nm thickness was added above and below the membrane resulting in ~100 water molecules per lipid in AA simulations and ~25 water beads per lipid in CG simulations. Systems were neutralized with K^+^ and Cl^−^ ions in 0.15 M concentration with the addition of neutralizing Ca^2+^ cations in the LPS core region in the case of Gram-negative outer membrane simulations. The peptide(s) were initially placed in solution in plane parallel with the membrane surface and ~2 nm above it, with hydrophilic and hydrophobic sides equally distanced from membrane surface. For the case AA-2, two bonded peptides from AA-12b simulations were taken as the initial peptide structure. In the CG-24 simulation, peptides were arranged in two parallel planes ~2 nm distant from each other. Two cases for AA-1 and AA-12 differed from each other in the positions of peptides. 

Equilibration was done in six steps, according to the CHARMM-GUI recommendations. Isothermal-isochoric (NVT) dynamics were used for the first two steps and NpT (constant pressure and temperature) dynamics were used for the other four steps. The temperature was fixed over the course of equilibration and production run at 310 K. During equilibration, various restraints were applied to the parts of the system: positional harmonic restraints to heavy atoms of the peptides, positional restraints on z-coordinates of lipid phosphorous atoms, and dihedral angle restraints applied on parts of lipids to prevent their unwanted structural change. These restraint forces were gradually reduced as the equilibration progressed [44].

Isothermal-isobaric (NpT) ensemble conditions were imposed by the Nose-Hoover thermostat and Parrinello–Rahman barostat, with a 1.0 ps time constant for the temperature and 5.0 ps for pressure (compressibility equal to 4.5 × 10^−5^ bar) [53,54]. The leapfrog integrator time step was fixed at 2 fs, and the bonds were handled by the LINCS option. The particle-mesh-Ewald method [55] was used for calculation of electrostatic interaction with coulomb cut-off at 1.2 nm and the van der Waals cut-off set to 1.2 nm with the force-switch at 1.0 nm.

The Gromacs utilities clustsize, traj and density were used for analysis of peptide aggregation, distances of hydrophobic/hydrophilic residues from membrane center, and density profiles along the membrane normal, respectively, while secondary structure was determined with the DSSP tool [56,57]. APL@Voro program, which is based on Voronoi partitioning of the lipid surface for selected key atoms in lipid headgroups, was used to calculate membrane thickness and area per lipid (APL) [58]. To determine the order parameters of lipid acyl chains, the Gromacs order utility was used.

Peptide amphipathicity was measured by the 2D-hydrophobic moment by HeliQuest [59], which uses the projection of a perfect helix on the 2D plane perpendicular to the central axis of the helix axis and Eisenberg scales to assign hydrophobicity to each amino acid residue (Appendix A). The 3D-HM tool [60] provided a more realistic three-dimensional hydrophobic moment (3D-HM) based on the charges assigned to atomic coordinates. The VMD program was used as a visualization tool [61]. Appendix A contains the majority of the analysis results, with the most important ones included in the article.

## 3. Results

We simulated interactions of one or more peptides with solvated membranes: (a) of POPE and POPG lipids, a model for a Gram-negative membrane; (b) of POPG, Lys-PG, and PVCL2 lipids, a model for a Gram-positive bacterial membrane; (c) of POPC lipids, a model for a eukaryotic membrane, and (d) a newly constructed model for an outer Gram-negative bacterial membrane (see Section 2).

The main focus was on exploring the molecular basis of adepantin-1 biological activity when it interacts with bacterial and eukaryotic cells. The putative mechanism of action was observed in three steps: (I) peptide initial contact and accumulation on the membrane surface; (II) peptide adaptation to the membrane’s polar and hydrophobic environment, and (III) peptide translocation and/or pore formation.

### 3.1. Accessing Peptide(s) Binding to Various Membrane with AA Modeling

As shown in Appendix A, one peptide is placed close to the membrane surface in the first simulation experiments. Adepantin-1 binds fast to Gram-positive and Gram-negative membranes but remains in solution when placed close to the POPC membrane (Appendix A). The peptide’s initial contact with the bacterial membranes is primarily due to electrostatic interactions as the charged side is oriented toward the polar surface of the membrane, and 3D-HM increases with the vector directed away from the membrane surface (Figure 1a). During the second step, the peptide position may fluctuate between partly bonded and unbonded states resulting in a favorable orientation of the 3D-HM vector for insertion (Figure 1b). In this step, the peptide rotates and unfolds, with the hydrophobic side moving closer to the membrane core, as evidenced by the change in 3D-HM vector orientation towards membrane interior (Figure 1c). The peptide, however, remains in the polar membrane region during the simulation time order of 1 μs, and the proposed final step in the adepantin-1 action mechanism is not observed.

These findings are also represented in Figure 2a,b, which shows the position of the center of mass of atoms (COM) from lysine (yellow lines) and hydrophobic (magenta lines) amino acid residues as a function of simulation time. During the first step, the polar side is close to the membrane (the yellow line is below the magenta line), and the peptide structure remains helical (Appendix A). The second step involves peptide adaptations to the membrane environment (both lines are intertwined), resulting in an unfolded structure with the hydrophobic side positioned deeper in the membrane (the yellow line is now on the top of the magenta line). The peptide interacting with both bacterial membranes exhibits a similar behavior, albeit with slightly different structure and dynamics. Figure 2c corresponds to peptide unfolding in water (Appendix A), in the case of the POPC membrane, characterized by larger fluctuations in COM distances and no partitioning of hydrophilic and hydrophobic residues. 

A similar two-step behavior is observed when multiple peptides are initially placed near bacterial membranes, as illustrated in Appendix A. Moreover, the binding process also included a very fast peptide association. Initially formed clusters have hydrophobic residues in contact with each other surrounded by hydrophilic residues that are exposed to water molecules or the membrane surface (Figure 3a and Appendix A). This cluster structure develops, where the hydrophilic residues in contact with the membrane surface shift, exposing some of the hydrophobic residues to the membrane environment (Appendix A). The clusters remain stable on the membrane surface, with mostly hydrophilic residues facing water molecules and some hydrophobic residues in contact with the lipid chains. Clustering is also observed in simulations of peptides with POPC lipids, where binding to the membrane is absent or rare, and clusters form in the water (see Appendix A). It is worth noting that the associated peptides, in water or in contact with the membrane, have a highly preserved α-helical structure (see DSSP plots in Appendix A).

The number of clusters as a function of time, presented in Figure 4a,b, varies significantly less for the peptides bound to Gram-positive than those bound to Gram-negative membrane, but mainly two clusters are formed in both cases at the end of simulation time. The same analysis for the clusters in water, observed in the simulations with the neutral membrane, shows a variable number of clusters containing two or more peptides (Figure 4c).

Cluster analysis showed that in almost all cases there is a high probability of the association of two peptides. Figure 5 shows that dimers are formed with a preserved initial amphipathic α-helical structure, with hydrophobic amino acids in close contact and with an anti-parallel conformation, where the C-terminal negatively charged Glu amino acid is located at opposite ends of the dimer. The AA-2 simulations demonstrated initial electrostatic contact and an adaptation process in which the dimer fluctuated between parallel and inclined positions with respect to the membrane surface. However, no transition from polar to hydrophobic contact with the membrane was observed, and the dimer remained on the membrane surface during the 1.5 μs simulation time.

### 3.2. Accessing Peptide(s) Binding to Gram-Negative Membrane with CG and CG2AA Modeling

To access longer simulation times and possible peptide insertion, we performed CG simulations with either twelve or twenty-four peptides interacting with the solvated Gram-negative cytoplasmic membrane using similar initial conditions to those in the AA simulations. Cases with varying peptide numbers were chosen to investigate the effect of different peptide to lipid ratios (P/L~1/64 and P/L~1/32) on the mechanism of action.

We concentrated on Gram-negative membrane because adepentin-1 is more effective against Gram-negative than Gram-positive bacteria and simulations of POPE:POPG bilayer are more computationally efficient. Further, we continued simulations by converting the final configurations of the CG models to AA models to obtain a more detailed description of interactions and potential peptide translocation across the membrane or membrane disruption. Appendix A depicts the simulation setup for the case with twelve peptides; additional results are available in Appendix A. 

During a 25 μs CG run, twelve peptides interacting with bacterial membranes showed similar clustering behavior to the atomistic simulations following the steps: (I) initial binding with a strong association of peptides, almost all of which were part of a single cluster with a hydrophobic core and a polar side in contact with the membrane surface; (II) adaptation, during which some peptides in the cluster reorganize exposing hydrophobic residues to the membrane’s hydrophobic region, and some of those spread out across the membrane surface as the simulation time progresses. Most of the peptides remained associated in one large cluster over the simulation time, and those that migrated mostly stay single or form two-peptide aggregates. It is noticeable that individual peptides, whether they are part of the aggregate or not, independently penetrate slightly deeper into the membrane staying close to the polar region. 

Similar behavior was observed in CG simulations with twenty-four peptides, with more significant membrane deformation caused by higher surface tension due to the higher P/L ratio of bound peptides (see Appendix A). 

In CG2AA simulations, some peptides, moved further into the membrane, below the membrane polar region, remaining nearly parallel to the membrane surface, with the hydrophobic side facing the hydrophobic membrane environment and preserving significant α-helical structuring, as shown in Figure 3b and Figure 5b. However, there appears to be no apparent membrane deformation or peptide translocation in either CG2AA simulations. 

Cluster analysis revealed that peptides preferentially form two-peptide clusters, remain as a single peptide, or are part of a larger cluster, supporting the AA simulation results. These findings are evidenced by the opposite correlation between the number of clusters and the maximum cluster size, which indicates the presence of one large cluster, and the bimodal character of the cluster size distributions (Figure 6), which is most noticeable in the case of CG2AA-12, where clusters are divided into two groups, one containing clusters with one and two peptides and the other with clusters of six and seven peptides.

### 3.3. Changes in Membranes Induced by Peptide Interaction

Several membrane properties defined by lipid composition have been put forward as contributing factors to AMP activity [1,62,63]. The polarity and charges of the surface region primarily contribute to the binding of peptides, which then further induces various structural and dynamical changes such as changes in curvature [64], thinning of the membrane [65,66] and destabilization of membrane integrity [22,67,68]. Here, the effect induced by peptides interactions with the membrane are described by area per lipid, membrane thickness and lipid order parameter.

Figure 7 shows, on the left, the ratio of area per lipid of the upper (with peptides) and lower leaflet of the membrane bilayer, and membrane thickness on the right for all AA and CG2AA simulation cases. Other results for this section are presented in Appendix A.

The main visible effect is a reduction in APL due to peptide insertion into the polar region of the upper leaflet, which is then followed by expansion of the lower leaflet. This decrease is smaller for the AA simulations but becomes more visible as the peptide-membrane interaction process is observed over longer times using the CG and CG2AA simulation settings. The results of CG2AA also support the fact that a higher P/L ratio causes a greater effect on APL; that is, with a higher P/L, the APL leaflets ratio decreases more. The results for Gram-negative and Gram-positive membranes differ when the time dependencies are observed, as shown in Appendix A, but the overall behavior is similar.

The results for membrane thickness follow a similar trend to those for the APL ratio, but with one exception, in that the thickness measured in AA simulations with twelve peptides is larger than in AA simulations with one or no peptide. This finding could be interpreted as an adaptation process in which the peptides pull the membrane and expand it, which is similar to the adaptation observed in the case of a single peptide (Figure 1b). The CG2AA simulations, on the other hand, revealed a significant decrease in membrane thickness, which decreases even more as the P/L ratio increases. These findings describe membrane thinning, which is commonly accepted as part of the mechanism of action of AMPs [68]. However, all observed changes in bacterial membranes are highly local and confined to the leaflet with the peptides, as exemplified in area of lipids and thickness profiles in Appendix A.

Membrane properties are also accessed by lipid order parameters, and here we show the parameters which measure the orientational mobility of the C–H bond for each lipid type, and each acyl chain averaged over the last 100 ns simulation time. Figure 8 presents results for POPG and POPC lipids while other results are in the Appendix A. The effect due to peptide-membrane interactions in AA simulations is small, but noticeable. In the case of the Gram-negative bilayer (Figure 8a,b) the binding of one peptide slightly reduces C-H bond mobility, while in interactions with twelve peptides there was no change or a slight increase in the order parameters compared to the case of membrane without the peptides. In the case of lipids of the Gram-positive membrane (Figure 8c,d), there is almost no difference in the order parameters when neither or some peptides are bonded to the membrane, and decrease when one peptide is in the membrane polar region. As expected, results for the POPC order parameters (Figure 8e,f), which also apply to other properties of the neutral membrane, are the same as those for the peptide-free membrane.

Similar to other membrane properties, a more significant effect was observed for CG2AA simulations, where peptide interaction with the membrane induced substantial increase in the orientational mobility of the C-H bonds of the acyl chain (Figure 8a,b, red and purple lines). We may interpret the overall results in that after initial peptide contact, the adaptation process (AA simulation) rigidifies the bacterial membrane, and subsequently, as shown in CG2AA simulations, makes it more fluid. Similar behavior has been reported for other AMPs [69].

### 3.4. Gram-Negative Outer Membrane Results

Antimicrobial peptides must interact and move across the cell wall of Gram-negative bacteria before they can potentially penetrate the inner membrane [70,71]. It has been proposed that permeabilization of the outer membrane occurs via a self-promoted mechanism [72]. According to this hypothesis, after initial binding of cationic AMPs to negatively charged LPS groups, AMPs compete with divalent cations, which non-covalently cross-link LPS molecules. Cationic AMPs eventually displace divalent cations because they have at least three orders of magnitude higher affinities for LPS and are much larger, disrupting the outer membrane locally. However, the detailed explanation of this process is still unknown [73].

In this work, the CHARMM-GUI [49] input generator was used to construct models of Gram-negative outer membranes that have two O-Antigen units, or zero O-Antigens connected to LPS molecules (see Section 2). As is known to the authors, this is the first time this type of membrane has been built, with its specific compositions modeling the outer bacterial membrane of *E. coli*. The simulations, cases with one and twelve peptides, were performed to access adepantin-1’s affinity for the various layers of the outer membrane. Additional data are available in Appendix A. 

In simulations of the smooth outer membrane (with 2 O-antigens), the peptide(s) rapidly bind to the outer membrane within a few ns (Figure 9 on the left) but stay mostly in the O-antigen region during the simulation time. On the other hand, adepantin-1 demonstrated low affinity for the rough outer membrane (without O-antigens) (Figure 9 on the right), as no, or some, peptide bindings were observed in the simulations. Therefore, a core region filled by divalent calcium cations may act as a barrier to peptide penetration of the OM membrane.

As reported in the literature, LPS has very slow dynamics; therefore, no significant insertion is expected to be observed during simulations on the order of nanoseconds [73]. Accordingly, simulations captured only the initial contact of adepantin-1 with the outer membrane but also indicated the differences in peptides interactions with the O-antigen and core region of the outer membrane. Further investigations will follow, including longer simulation times and using advanced simulation methods needed to investigate the characteristics of mechanisms by which peptides pass through the outer membrane layers.

## 4. Discussion

Simulations have proven to be extremely useful in uncovering functionality of membrane-active peptides with atomistic details [74,75,76]. The continuous increase in computer power brought about by the efficient use of GPUs, as well as the development of accurate atomistic and coarse-grained models, has accelerated the transition from simulations of simplified model membranes to simulations of multicomponent realistic membranes [23]. Furthermore, the use of simulations is well supported by the available programs for system construction and post-simulation analysis. Here, we performed AA and CG combined with AA simulations, with the initial conditions as close as possible to the potential binding process, including the α-helical structure that the peptide is expected to conform to after binding. Each of the specific simulation strategies was aimed at observing the specific progress in adepantin-1’s mechanism of action.

The simulation results demonstrated that adepantin-1 binds strongly to both types of cytoplasmic membranes, a Gram-negative membrane modeled after *E. coli* and a Gram-positive membrane modeled after *S. aureus*. Small differences in peptide structure and dynamics distinguished interactions with Gram-positive and Gram-negative cases, but these differences were insufficient to explain the higher antibacterial activity observed in *E. coli* MIC compared to *S. aureus* MIC measurements. Therefore, it can be assumed that interactions with components of the extracellular medium and bacterial outer membrane and/or peptidoglycan layers [71,77], may be important in defining the spectrum of activity and selectivity of adepantin-1 against Gram-negative bacteria. As indicated by the results of outer membrane simulations, adepentin-1 has different mechanisms of passage through the membrane layers. The simulations confirmed that dimerization and multi-peptide associations are important components of adepantin-1 activity, as suggested by CD spectra analysis of adepantin-1 interaction with anionic LUVs and as well as the higher biological activity observed for covalently bound dimers [32]. The observed low hemolysis of adepentin-1 is also well represented by the simulation results, as simulation of adepentin-1 with a POPC membrane (eukaryotic cell model) revealed little or no binding of the peptides. Moreover, there is correspondence with the measurements of adepantin-1 CD spectra in the presence of neutral LUVs, as they both show peptide structure destabilization, which, however, in simulations, is mostly observed for non-associated peptides. 

However, the possible membrane-related mechanism of adepentin-1 biological activity was only partly captured by the simulations, and the observed results are primarily related to the molecular details of binding and insertion processes. Adepantin-1, as a single, dimer or multiple peptides, interacts with the bacterial membrane with initial electrostatic contact with the polar membrane surface followed by insertion with the hydrophobic residues in contact with hydrophobic membrane environment. Similar behavior has been observed for the antimicrobial peptide PGLa [78], including reorientation and dimerization. The main feature of the adepentin-1 molecular mechanism is the high affinity for associations. Clusters are formed in solution or simultaneously with peptide binding to the membrane surface. These aggregates are primarily due to hydrophobic forces, as evidenced by the cluster’s hydrophobic core and polar surface. Moreover, the associated peptides preserve mostly the initial α-helical structuring, thus indicating that aggregates may act as folding promotors. 

The effect of peptide aggregation on antimicrobial activity is still debated, and it most probably depends on the peptide type. For example, in a seminal paper, Sengupta et al. showed that aggregations at or near the membrane provide the critical local concentration needed for AMP activity achieved by pore formations [11]. In a recent study [79] it is argued that peptide self-associations exert antimicrobial activity by providing an amphipathic environment that allows them to adopt a helical structure. A contrary view is presented by Zou et al. who showed that antimicrobial activity for both guanine-modified magainin II and cecropin A-melittin decreases with increased peptide self-aggregation, which contributes to the increased energy cost of the peptide embedding into the cell membrane [80].

Here, it was observed that peptides initially associated in one large cluster slowly move away and diffuse in the membrane polar region, mostly staying as a single peptide or associating by pairs forming an antiparallel dimer. We can also assume that aggregation facilitates peptide placement in a favorable conformation for insertion, with α-helical structuring and the hydrophobic side in contact with the membrane’s hydrophobic core.

Therefore, based on the observed interplay between polar and hydrophobic contributions, we propose the following mechanism. At the core of adepantin-1’s behavior is its amphipathic α-helical structure, which was implemented in its design. Amphipacity is a well-known feature of AMPs, but it is mainly discussed as a favorable property for adapting a single peptide to the polar and hydrophobic environment of the membrane [81,82]. Here we focused on peptide self-associations promoted by the amphipathic structure. In a polar environment, in water or at the membrane surface, peptide self-association is more favorable, and aggregates are stable with the polar surface and the hydrophobic amino acids buried inside. However, when the aggregate moves further into the membrane, coming into contact with the hydrophobic environment, hydrophobic residues are exposed at the interface, and the stability of the aggregate decreases leading to separation of the peptides from the aggregate.

However, peptide translocation or membrane disruption were not observed in either all-atom or coarse-grained simulations, so we can speculate that higher peptide concentrations may cause greater membrane deformations, potentially leading to membrane disruption [68]. It is worth noting that according to the literature, the CHARM force field makes a significant contribution to the high stability in membrane and peptide-membrane systems, and translocation or pore formation is unlikely, in contrast to simulations with the OPLS force field, where membrane instabilities in interaction with AMPs are more likely to be observed [68,78].

Finally, efforts to develop drugs based on host defense peptides typically involve fine-tuning various structural aspects that contribute to activity with the goal of increasing bactericidal activity while decreasing toxicity [19]. A variety of studies are also required to develop a more general picture of specific structural and dynamical aspects influencing potency and selectivity. As this and other studies show, it might also be necessary to consider the effect of self-associations in the design of AMPs [80]. Here, the presented results corroborate well the rules implemented in AMP-Designer, as designed amphipathic sequence strongly defines adepantin-1 molecular mechanisms. Additionally, as shown in Figure 10, the positions of smaller (Gly and Ala) and larger (Leu) amino acids in adepantin-1 sequence allow for the formation of closer or more distant hydrophobic contacts, and amino acid Glu at the C-terminus contributes to the formation of an antiparallel dimer. 

Research is ongoing, and in a recent study, adepantin-1 was modified with three amino acid substitutions to gain even broader spectrum activity against Gram-positive and Gram-negative bacteria while maintaining selectivity and low toxicity to healthy human cells [83]. Substitutions were chosen with the Mutator tool [84] and the adepantin-1 analog GI**K**K**A**VGKALKGLKGLLK**A**LGES-NH_2_ (substituted residues are in bold font) showed improved antibacterial activity compared to adepantin-1. Moreover, adepentin-2 GIGKHVGKALKGLKGLLKGLGE**C**–NH_2_ with Ser to Cys modification [32] was enlisted as one of the most promising drug candidates in a recent review [85].

## 5. Conclusions

Antimicrobial peptides are among the molecules being investigated as potential antibiotics. Finding or designing AMPs with potent antibacterial activity and low toxicity to human cells is a difficult task that necessitates a thorough understanding of peptide-membrane interactions at the molecular level. Self-associations of AMPs may also play an important role in their mechanism of action and, therefore, it is worth including the information on peptide-peptide interactions in design algorithms.

More research such as that presented in this paper is needed to observe peptide activity when interacting with different types of membranes in order to reveal the specificity of lipid composition contributions, thereby propelling the design of novel antimicrobial peptides aimed at clinical applications.

## Figures and Tables

**Figure 1 membranes-12-00891-f001:**
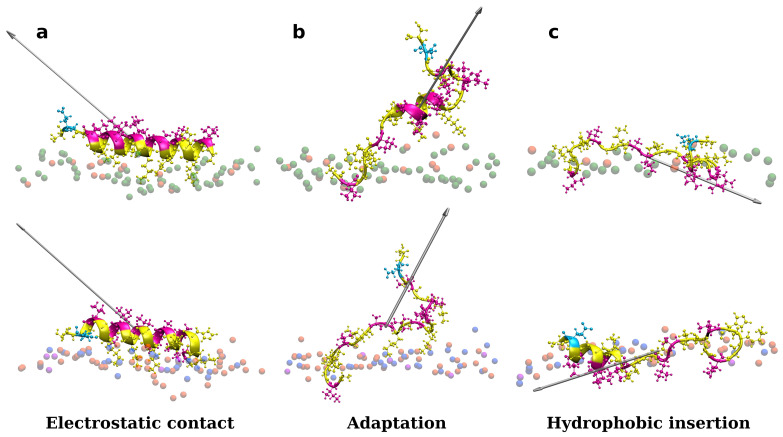
Representations of a single adeptantin-1 interacting with Gram-negative (on the top) and Gram-positive (on the bottom) bacterial membranes, where snapshot (**a**) shows electrostatic bonding of charged residues with upper leaflet lipids, (**b**) depicts peptide conformation change, and (**c**) represents the final state with hydrophobic residues inserted in the membrane interior. The 3D-HM vector is depicted as a grayed arrow, where length of the arrow corresponds to the 3D-HM value (values are in Appendix A). Peptides are depicted as ribbons and spheres, with hydrophobic residues in magenta, polar and positively charged residues in yellow, and negative Glu in cyan. The upper leaflet of the membrane is represented by beads for P atoms that are shown in green for POPE lipids, orange for POPG lipids, blue for Lys-PG lipids, and violet for PVCL2 lipids. For clarity, other atoms and molecules of the membranes, as well as water molecules and ions, are removed.

**Figure 2 membranes-12-00891-f002:**
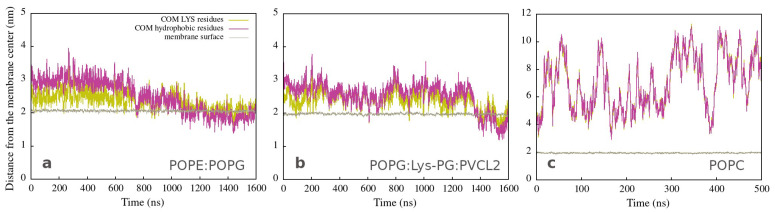
Distances from the membrane center to the COM of LYS (yellow line) and of hydrophobic (magenta line) residues as a function of time for simulations of a single peptide with (**a**) Gram-negative, (**b**) Gram-positive, and (**c**) neutral membranes. The brown line represents the average position of the membrane surface, which is defined as half the membrane thickness (calculated by APL@Voro).

**Figure 3 membranes-12-00891-f003:**
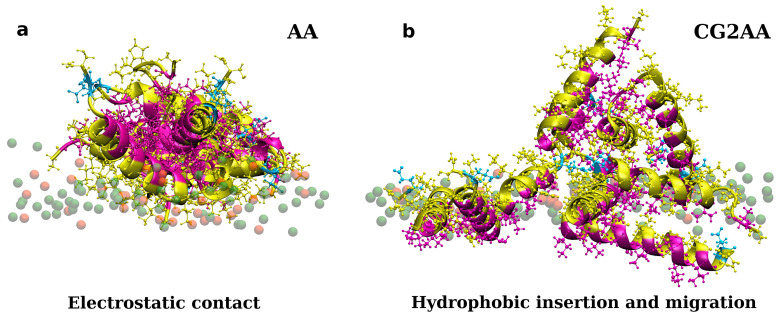
Representations of twelve adepantin-1 peptides interacting with the Gram-negative bacterial membrane. Snapshot (**a**) is the AA-12 simulation result that shows electrostatic bonding of charged residues with upper leaflet lipids. Snapshot (**b**) is the result of CG2AA simulation, depicting further steps in peptide-membrane interactions in which peptides associated when binding to the membrane separate and migrate along the membrane surface with hydrophobic residues inserted in the membrane interior. Peptides are depicted as ribbons and spheres, with hydrophobic residues in magenta, polar and positively charged residues in yellow and negative Glu in cyan. The upper leaflet of the membrane is represented by beads for P atoms that are shown in green for POPE lipids. For clarity, other atoms and molecules of the membranes, as well as water molecules and ions, are removed.

**Figure 4 membranes-12-00891-f004:**
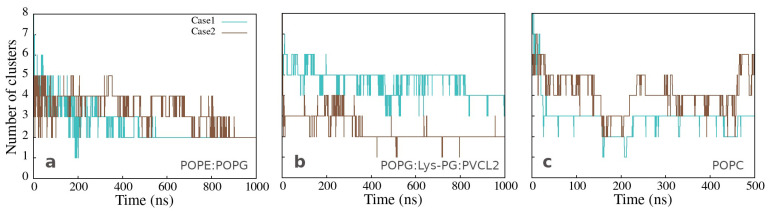
Number of clusters as a function of simulation time for simulations of twelve peptides with (**a**) Gram-negative, (**b**) Gram-positive, and (**c**) neutral membranes. The calculation was done with the clustsize Gromacs utility, with the condition that a peptide belongs to a specific cluster if the distance between its atom and any atom in the cluster is less than 0.35 nm. Other clusters properties such as maximum cluster size, cluster size distribution, and representative cluster snapshots are provided in Appendix A.

**Figure 5 membranes-12-00891-f005:**
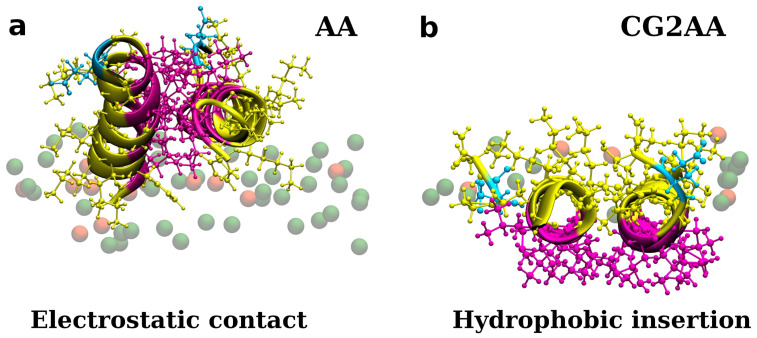
Representations of two adepantin-1 peptides interacting with Gram-negative bacterial membrane. Snapshot (**a**) is the AA-2 simulation results that show electrostatic bonding of charged residues with upper leaflet lipids. Snapshot (**b**) is the result of CG2AA simulation, depicting further steps in membrane-peptides interactions, where dimer is rotated placing hydrophobic residue in contact with the membrane hydrophobic core. Peptides are depicted as ribbons and spheres, with hydrophobic residues in magenta, polar and positively charged residues in yellow and negative Glu in cyan. The upper leaflet of the membrane is represented by beads for P atoms that are shown in green for POPE lipids. For clarity, other atoms and molecules of the membranes, as well as water molecules and ions, are removed.

**Figure 6 membranes-12-00891-f006:**
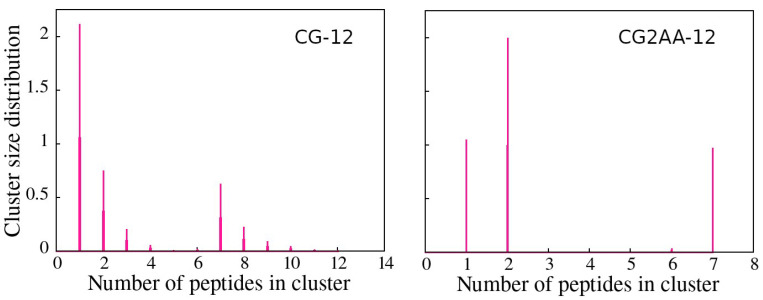
Cluster size distribution as a function of the number of peptides in a cluster, with the CG-12 simulation results on the (**left**) and the CG2AA-12 simulation results on the (**right**). The calculation was done with the clustsize Gromacs utility, with the condition that a peptide belongs to a specific cluster if the distance between its atom and any atom in the cluster is less than 0.35 nm. Other clusters properties, such as maximum cluster size, cluster size distribution, and representative cluster snapshots are provided in Appendix A.

**Figure 7 membranes-12-00891-f007:**
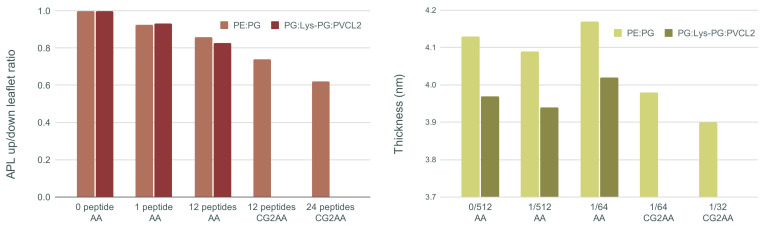
On the (**left**) ratio of upper and lower leaflet area per lipid (APL) averaged over the last 100 ns of simulation time, and on the (**right**) membrane thickness averaged over the last 100 ns of simulation time for each case of AA and CG2AA simulations. Calculations are done with the APL@voro tool [58].

**Figure 8 membranes-12-00891-f008:**
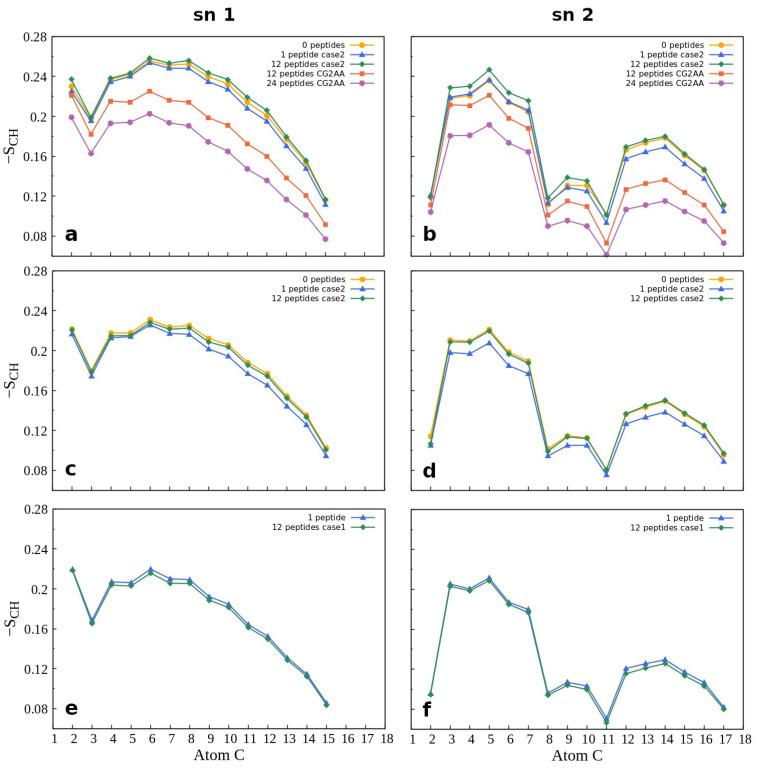
Order parameter −S_CH_ for the acyl chains sn-1 (on the left) and sn-2 (on the right) as a function of the carbon atom index. The first row (**a**,**b**) shows the results for POPG lipids in POPE:POPG membrane, while the second row (**c**,**d**) shows the calculations for POPG lipids in POPG:Lys-PG:PVCL2 membrane, and the third row (**e**,**f**) shows calculations for POPC lipids in POPC membrane. Appendix A shows additional order parameter calculations. For calculation, the order Gromacs utility was used.

**Figure 9 membranes-12-00891-f009:**
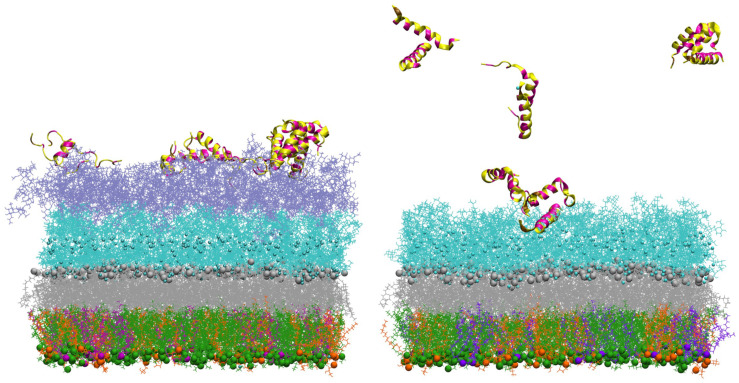
Representations of the Gram-negative bacteria’s outer membrane interacting with twelve peptides initially placed in solution 2 nm above the membrane’s last layer. On the (**left**), snapshot of the outer membrane with 2 O-antigen units at 400 ns simulation time, and on the (**right**), snapshot of the outer membrane at ~100 ns simulation time. PVPE lipids are green, PVPG lipids are orange, PVCL2 is violet, and Lipid A is gray, all represented by sticks and beads. The core region is represented by cyan lines, neutralizing Ca^2+^ ions in core region as small cyan beads and the 2 O-antigen units are represented by purple lines. Peptides are shown as ribbons, with polar and charged residues in yellow and hydrophobic residues in magenta. Water molecules are not shown for clarity.

**Figure 10 membranes-12-00891-f010:**
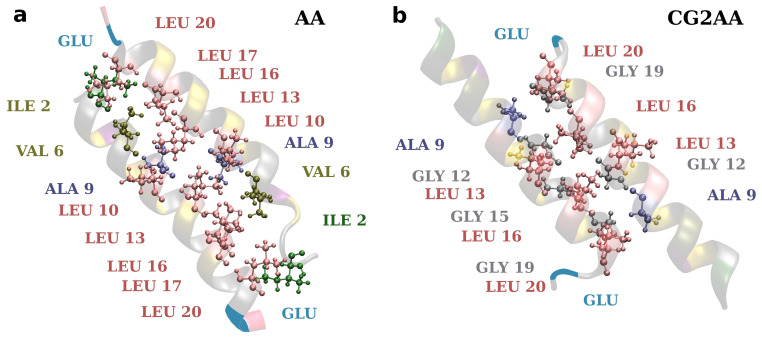
Representations of hydrophobic contacts in two-peptide association. The snapshot on the left is from the AA-2 simulation and depicts dimer contacts when it is electrostatically bonded to the membrane surface (corresponding to Figure 5a). The snapshot on the right is from the CG2AA simulation representing contacts when the dimer is deeper in the membrane (corresponds to Figure 5b). The distance between the peptide axis is smaller (~0.75 nm) in case (**b**) than in case (**a**) (~0.90 nm). Ribbon models with colored amino acids depict the peptides, and amino acids participating in hydrophobic contacts are specified. The Glu is also added to depict dimer antiparallel configuration. The criterion for hydrophobic contact is that any atom from one residue is within 0.35 nm of any atom in another residue.

**Table 1 membranes-12-00891-t001:** Summary of the MD simulations.

Label	No. of Peptides	Gram −		POPC		Gram +		Gram − Outer Membrane		
		Time $\left(\mu s\right)$	No. of Lipids	Time $\left(\mu s\right)$	No. of Lipids	Time $\left(\mu s\right)$	No. of. Lipids	2 O-Antigens	0 O-Antigens	No. of Lipids and LPSs
								Time $\left(\mu s\right)$	Time $\left(\mu s\right)$	
AA-0	0	0.5	192 POPE 64 POPG	-	256 POPC	0.5	146 POPG 96 Lys-PG 14 PVCL2	0.5	0.5	Up: 50 LPS Down: 105 PVPE 30 PVPG 8 PVCL2
AA-1 a *	1	1		0.5		1		0.5	0.1	
AA-1 b *	1	1.6		-		1.6		-	-	
AA-2	2	1.5		-		1.5		-	-	
AA-12 a *	12	1	384 POPE 128 POPG	0.5	512 POPC	1	292 POPG 192 Lys-PG 28 PVCL2	-	0.1	
AA-12 b *	12	1		0.5		1		0.4	0.2	
CG-12	12	25	384 POPE 128 POPG							
CG-24	24	42.5								
CG2AA-12 **	12	0.5								
CG2AA-24 **	24	0.5								

* Case1 simulations are marked with the letter a, and case2 with the letter b. ** Last frame of CG simulations was transformed to atomistic model, and simulation continued as AA MD.

## Data Availability

All simulation data are available by sending a request to the authors.

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
