# Peer review of "Simulation Study of the Effect of Antimicrobial Peptide Associations on the Mechanism of Action with Bacterial and Eukaryotic Membranes"

_membranes, 2022, doi:10.3390/membranes12090891_

Round 1

Reviewer 1 Report

Manuscript of Matko Maleš and Larisa Zoranić is devoted to investigation of the mechanism of interaction of antimicrobial peptide (AMP) adepantin-1 with model membranes.

The authors have done a lot of computational work, modeled sufficiently long trajectories using CG and AA MD modeling. Four different model membranes were used and an influence of the peptide/lipid ratio was also studied. An additional advantage of the work is the use of more complex and realistic models of Gram+ bacteria and model of the outer membrane of E. Coli. 

Authors declare that “By using simulation experiments, we aim to uncover a molecular basis for the experimentally observed results.” In my opinion, they have not achieved their goal. Despite the abundance of material, the authors did not show any new interesting results. They only confirmed that cationic peptides interact with anionic lipids and exhibit weak interaction with zwitterionic lipids. There is not enough detailed analysis of the results obtained. The section “3.2.4. Gram-negative outer membrane results” looks unfinished and incomplete.

Several additional remarks:

1.      Some information is missing in the Materials and Methods section:

What is the size of the model membranes? How many and what lipids were in each leaflet?

How were the area per lipid and the thickness of bilayers calculated?

2.      The section “3.1. Peptide design and biological activity” does not contain the results of this work and should be moved to the Materials and Methods or Introduction.

3.      I would recommend to renumber and to move the figures in Supplementary according to their discussion in the text. Some figures (for example, S13) demand additional explanations.

4.      Order parameters in fig. 7a-b reduce in CG simulations showing an increase in the mobility of CH bonds.

Round 2

Reviewer 1 Report

I can recommend the manuscript for publication.

Reviewer 2 Report

I am mostly satisfied with the corrections, but in lines 138 and 139, the authors should write the names of the cells in italic, please.